# Molecular Dynamics Simulations Reveal Canonical Conformations in Different pMHC/TCR Interactions

**DOI:** 10.3390/cells9040942

**Published:** 2020-04-10

**Authors:** Josephine Alba, Lorenzo Di Rienzo, Edoardo Milanetti, Oreste Acuto, Marco D’Abramo

**Affiliations:** 1Department of Chemistry, University of Rome Sapienza, P.le A.Moro 5-00185 Rome, Italy; 2Department of Physics, University of Rome Sapienza, 5-00185 Rome Italy; lorenzo.dirienzo@uniroma1.it (L.D.R.); edoardo.milanetti@gmail.com (E.M.); 3Center for Life Nano Science@Sapienza, Istituto Italiano di Tecnologia, 000161 Rome, Italy; 4Sir William Dunn School of Pathology, University of Oxford, Oxford OX1 3RE, UK; oreste.acuto@path.ox.ac.uk

**Keywords:** molecular dynamics, biophysics, protein-membrane, T cell antigen receptor

## Abstract

The major defense system against microbial pathogens in vertebrates is the adaptive immune response and represents an effective mechanism in cancer surveillance. T cells represent an essential component of this complex system. They can recognize myriads of antigens as short peptides (p) originated from the intracellular degradation of foreign proteins presented by major histocompatibility complex (MHC) proteins. The clonotypic T-cell antigen receptor (TCR) is specialized in recognizing pMHC and triggering T cells immune response. It is still unclear how TCR engagement to pMHC is translated into the intracellular signal that initiates T-cell immune response. Some work has suggested the possibility that pMHC binding induces in the TCR conformational changes transmitted to its companion CD3 subunits that govern signaling. The conformational changes would promote phosphorylation of the CD3 complex ζ chain that initiates signal propagation intracellularly. Here, we used all-atom molecular dynamics simulations (MDs) of 500 ns to analyze the conformational behavior of three TCRs (1G4, ILA1 and ILA1α1β1) interacting with the same MHC class I (HLA-A*02:01) bound to different peptides, and modelled in the presence of a lipid bilayer. Our data suggest a correlation between the conformations explored by the β-chain constant regions and the T-cell response experimentally determined. In particular, independently by the TCR type involved in the interaction, the TCR activation seems to be linked to a specific zone of the conformational space explored by the β-chain constant region. Moreover, TCR ligation restricts the conformational space the MHC class I groove.

## 1. Introduction

Cytotoxic T cells recognize and kill virus-infected cells and cancer cells upon T-cell antigen receptor (TCR) interaction with major histocompatibility complex (MHC) class I proteins presenting viral and tumor antigens, respectively [1,2,3]. MHCs class I are composed of one α-chains, which form the binding site for a nine/ten residue-long peptide, and of a non-covalent bound β2 microglobulin (Figure 1). To recognize pMHC, T cells use a clonally distributed αβ dimer with Ig-like variable domains, Vα and Vβ. Together, Vα and Vβ form the pMHC binding site composed of six loops homologous to antibody complementarity determining regions (CDRs) 1, 2 and 3 [4,5]. CDR1 and CDR2 have limited variability, while CDR3s are hypervariable. VαVβ orientates diagonally relative to the long axis of the peptide-binding groove [4,5] in such a way that the CDR3s make contacts mostly with the peptide whereas the CDR1s and CDR2s contact mainly the MHC. Vα and Vβ connect to Ig-like constant domains (Cα and Cβ) that are linked to transmembrane regions (TMRs) via a connecting stalk (CP). pMHC binding is signaled intracellularly by four non-covalently associated subunits (γ, δ, ε and ζ), called CD3, that are organized into three dimers (γε, δε, ζζ) [6]. δε and γε ectodomains interface with Cα and Cβ ectodomains, respectively, while ζζ has a very short ectodomain. All CD3 subunits possess short and unstructured intracellular tails that are phosphorylated soon after pMHC binding at the immunoreceptor tyrosine-based activation motifs (ITAMs). Early biochemical work [7] suggested TCR-CD3 allosteric regulation and fluorescence-based studies has suggested a movement of a Cα loop upon pMHC binding. Moreover, recent NMR investigations of soluble αβ dimer ectodomain alone has suggested that TCR-CD3 signaling is governed by allosteric regulation upon pMHC binding [8,9].

Combining experimental and theoretical studies, we recently found that the bound peptide can affect the conformation of the MHC I binding groove, suggesting a different presentation of the antigens, which seems to be related to different CTLs responses [10]. However, in that study we did not provide a modelling of the pMHC interacting with the TCR. To better characterize such a complex molecular network, here we present the first exhaustive computational study of five pMHC-TCR complexes, modelled in a heterogeneous lipid bilayer (see Appendix A, and Figure 1).

## 2. Materials and Methods

### 2.1. Complexes Modelling 

Using the crystallographic structures (PDB ID: 4MNQ; PDB ID: 2BNR) the transmembrane alpha helices were built by means of the Modeller Software version 9.19 (Accelerys, San Diego, CA, USA) [11], following the amino acid sequences provided by the Uniprot database [12] (Uniprot entries P01848 and P01850 for the TCR α and β chains respectively; Q9MY51 for the α chain of the HLA-A* 02:01). The heterogeneous lipid bilayer was built by means of the CHARMM-GUI Membrane Builder web software (Harvard, Cambridge, MA, USA) [13], with a composition of: 1-palmitoyl-2-oleoyl-sn-glycero-3-phosphocholine (POPC) at 90%, phosphatidylinositol (4,5)-bisphosphate (PIP2) at 7% and 1-palmitoyl-2-oleoyl-sn-glycero-3-phospho-L-serine (POPS) at 3%. Such a lipid composition is based on the work of Chavent et al. [14] and Zech et al. [15] to approximate the lipid composition of a mammalian cell membrane [16]. These studies reported an essential role of the PIP2 in receptor activation and TCR triggering. Note that recent findings reported that cholesterol can inhibit T cell activation and thus, this lipid was not included in our membrane model [17,18]. The modelled complexes were then manually inserted in the membrane using the VMD Software version 1.9.3 [19] (University of Ilinois, Champaign, IL, USA).

### 2.2. MD Simulations

The systems were solvated with the TIP3P water model [20] and neutralized with Na+ and Cl- ions at physiological concentration (0.15 M). The exceeding solvent was manually removed, excluding the water molecules within a range of 2 Å from lipids. An energy minimization step was performed using the steepest descent algorithm without position restraints. After the minimization, a series of equilibration steps were performed: (1) a NPT equilibration of 40 ps was run to allow the packing of the lipids around the protein, using an integration time step of 0.2 fs; then the NPT equilibration was extended until 2 ns. (2) an NVT equilibration of 40 ps was performed and then extended until 4 ns, increasing the time step at 1 fs. (3) A latest NPT simulation 500 ns long was run with a time step of 2 fs. The V-rescale thermostat [21] and the Parrinello-Rahman barostat [22] were used with a τT = 0.1 ps and a semi-isotropic coupling with a τp = 5 ps, respectively. The temperature, considering the melting point of the lipid composition, was kept constant at 305 K. The electrostatic interactions were calculated using the particle mesh Ewald method [23] with a cut-off of 1.2 nm. A cut off of 1.2 nm was used for the van der Waals interactions. The simulations were performed using the additive for lipids CHARMM-36 force field [24,25], and the Gromacs Software version 2018.1 (University of Groningen, Groningen, Netherlands) [26]. For each system, we performed a Molecular Dynamics simulation lasting 500 ns, resulting in an aggregated time of four microseconds.

### 2.3. Structural Analysis

The root mean square deviation (RMSD) is a statistical measure of the average distance between a selected group of atoms, with respect to a reference structure. According to the following equation (Equation (1)):(1)$$\mathrm{RMSD}=\sqrt{\frac{1}{N}\sum_{i=1}^{N}{\left|r_{i}\left(t\right)-r_{i}^{0}\right|}^{2}}$$
where *r_i_ (t)* is the position of the atom *i* at the time *t*, N is the total number of atoms in the group considered and *r_i_^0^* is the position of atom *i* in the reference structure. The RMSD calculation was performed considering the alpha carbons, choosing the first frame of the simulation as reference.

The root mean square fluctuation (RMSF) is a statistical measure of the deviation between the position of the atom *i* (or a group of atoms, e.g., a residue) *r_i_ (t)*, and the initial structure *r_i_^0^* considering the time interval T (Equation (2)):(2)$${\mathrm{RMSF}}_{i}=\sqrt{\frac{1}{T}\sum_{t_{j}=1}^{T}{\left|r_{i}\left(t_{j}\right)-r_{i}^{0}\right|}^{2}}$$

Such a measure allows to detect and quantify the displacement of the different protein regions along the MD simulation.

### 2.4. Essential Dynamics

The essential dynamics technique is a statistical method based on the principal component analysis [27]. Briefly, the covariance matrix of the atomic positions is built from the MD simulations on a selected group of atoms (usually C-alpha). From the diagonalization of such a matrix, a set of eigenvectors and associated eigenvalues is obtained. The eigenvectors represent the principal motion directions of the system and, therefore, they are used to describe the “essential” protein modes, which often represent the functional ones. In this way, the fastest motions present in the simulations, which describe biologically not relevant motions (i.e., vibrations), are excluded making possible to represent the protein dynamics in a reduced space-as defined by the eigenvectors—which approximate well the overall molecular motions.

The essential subspace, describing the overall motion, is mostly confined within the first 2 eigenvectors, in the case of study. Combining two (or more) trajectories of different systems (having equal alpha carbons numbers) it is possible to obtain common eigenvectors defining the subspace explored by the different proteins. The projections of the MD trajectory on the first 2 eigenvectors (i.e., principal components), allow the comparison of the conformations assumed by the proteins during the simulation. We compared the conformational behavior of the pMHC-TCR systems in study, analyzing the entire complex and then the single regions.

The gmx covar and gmx anaeig tools of the Gromacs Software 2018.1 [26] were used to build the covariance matrix and to calculate the 2d projections with respect to the first 2 eigenvectors.

### 2.5. Cross-Correlation Matrix

The cross-correlation is the correlation between the entries of two random vectors X and Y, while the correlations of a random vector X are the correlations between the entries of X itself, those forming the correlation matrix of X. In such a matrix, the correlations of the various temporal instances of X with itself are known as autocorrelations, and they are arranged on the matrix diagonal. Outside the diagonal, there are the cross-correlations between X and Y across the time, which assume the value between +1 and −1. We considered that the regions are correlated when such a value is greater than 0.75, and they are anti-correlated from −1 to −0.25. The cross-correlation matrix was computed by means of the Bio3d package of the R Software version 3.5.3 [28,29] (University of Michigan, Ann Arbor, MI, USA).

### 2.6. Construction of the Zernike Descriptor

For each MD frame we calculated the molecular surface and the electrostatic potential by means of PDB2PQR [30] and Bluues [31] software package. Then, we extracted by means of a voxelization procedure the three Zernike 3D functions (3DZD) [32,33], representing the shape, the positive electrostatics and the negative electrostatics of the selected region, i.e., the binding groove. Such a procedure was recently implemented and applied in our recent work on similar systems [34,35].

### 2.7. Network Analysis

To investigate the topological and structural properties of the different systems, we have adopted a graph theory approach. To this end, we have selected about 100 frames for each simulation and each structure has been represented as a network, where each residue is a node (or vertex) of the graph. Two nodes are joined by a link (or edge) if the distance between the alpha carbon of the corresponding residues is lower than a threshold (6 Å). To analyze the contribution of each residue within the protein network, we have considered two local descriptors: “closeness centrality” and “degree”. The “closeness centrality” of a node is a local network parameter, measuring the mean number of steps required to reach any other protein residue (or node), starting from that node. The node “degree” is one of the most used topological descriptors in graph-theory based approach, counting the number of links of that node (in other words the node degree is the number of residues closer than the cutoff to the examined residue). To highlight the network characteristics from a structural point of view, each residue of the complex was colored in accordance with the corresponding value of the network parameter. For both the network parameters, a boxplot was used 164 to represent the distribution of the values for each residue [36].

### 2.8. Membrane Analysis

The membrane thickness was computed by means of the GridMAT-MD program [37]. Such a tool splits the lipid bilayer into a grid. To obtain a better resolution, a grid of 100 × 100 points was selected to perform the analysis. The distance between the two bilayers was calculated using the gmx mindist tool of the Gromacs Software [26].

## 3. Results

Due to the availability of kinetic data present in the literature [38,39,40,41,42,43], we modelled the HLA-A*02:01 containing the peptide ESO9C (the tumor antigen NY-ESO157-165 fragment—SLLMWITQC [40,42]), the mutated ESO4D (a.a. sequence SLLDWITQV [42]), the hERT540-548 antigen (a human telomerase reverse transcriptase epitope-sequence ILAKFLHWL [38,41,43]) and the mutated analogous (a.a. sequence ILAKFAHWL, named below as hTERT-6A [41]). The complexes with the NY-ESO-derived peptides were modelled in presence of the 1G4 TCR, while the hTERT derivatives with the ILA1 TCR and the ILA1α1β1. This latter TCR is mutated in the CDR2β and CDR3α loops, as shown in Table 1. In addition, the A*02:01-ESO9C, the 1G4 and ILA1α1β1 were separately simulated in a membrane environment, as reference. The simulation of the complexes in a membrane environment makes it possible to study the effect of the lipidic phase on the pMHC-TCR dynamical behavior, which was recently indicated to play an important role in the receptor activation [44].

### 3.1. Conformational Analysis

Due to the complexity of the simulated systems, the RMSDs of the pMHC-TCR complexes reach a plateau after several ns (Appendix A), and, thus the first 50 ns were removed from the analysis. To understand the complex dynamics behavior of the simulated systems, we first looked at the RMSF analysis, which provides a simple representation of the fluctuation behavior (Figure 2). Such an analysis shows that the more fluctuating regions are almost conserved, despite relevant differences in the sequences between the simulated systems (Appendix A). A more detailed analysis of the RMSF on specific regions, points out that the 1G4 construct is more affected by the peptide changes, especially in the Vβ region (Appendix A).

#### 3.1.1. Collective Motions

The overall motions of the simulated systems have been compared by means of the Essential Dynamics method. Such an analysis shows that a large part of the motions can be represented by the first few principal eigenvectors. In fact, the first two eigenvectors describe about 50% of the system total variance, making possible to describe the main conformational behavior in the subspace described by the first two eigenvectors. Therefore, we firstly projected the conformations explored by the bound and unbound states of the alpha carbon of the 1G4 and the ILA1α1β1 complexes and then all the bound simulations were compared to each other.

#### 3.1.2. Comparison between the Bound and Unbound States

Combining the MD trajectories of the 1G4 systems, we performed the analysis on the TCR alpha carbons, and, separately, on the variable and constant regions. As shown in Appendix A, no large conformational rearrangements occur in the time range of the MD simulations, and the overall motions of the bound form of the 1G4 TCR are comparable with respect to the unbound one. On the contrary, the Vβ regions of the bound systems project on different regions with respect to the unbound one (Figure 3a,b). Here the CDR2β and CDR3β regions are the movement-determining sites. This is in line with the work of Housset et al. [45], where a significant conformational change of the CDR3β loop, necessary for the binding to the pMHC was found. Interestingly, the Cα region projections show a larger movement of such a region in the bound state, enclosing conformational rearrangements, which are different for the 1G4-ESO9C and ESO4D (Appendix A). This observation seems in line with the conformational change previously observed in the Cα region of the TCR by means of fluorescence spectroscopy [46]. Even more interesting, the Cβ region projections are quite overlapped in the essential subspace, suggesting a similar conformational behavior of the TCR (Figure 3). The projections of the TCR conformations sampled by the ILA1α1β1 simulations (Appendix A), show that such an unbound state explores different conformation with respect to the corresponding bound systems, e.g., ILA1α1β1-hTERT and hTERT6A. The analysis of the variable regions underlines the presence of different conformations of the Vα between the bound and the unbound state (Appendix A), and even more in detail, a conformational change occurs in the ILA1α1β1-hTERT. On the contrary, the Vβ conformations explore the same region of the conformational space (Figure 3c). Finally, the constant region projections discriminate between the bound and unbound ILA1α1β1 TCR (Figure 3d).

#### 3.1.3. Comparison between the Bound States

The essential dynamics of the five pMHC-TCR complexes show that the first eigenvector is able to discriminate between the two different TCR types. Noteworthy, such an eigenvector principally describes the motions of the variable and the constant regions, suggesting that differences in TCR overall motions are essentially due to the dynamical behavior of such regions (Figure 4a and Appendix A). Interesting, all the Cβ regions conformations explore similar areas on the essential subspace, except for the ILA1 which has no binding affinity for the HLA-A*0201:hTERT-6A (Figure 5d). Finally, the binding groove alpha carbons projections (residues 1-175 of the HLA-A*0201 alpha chain) reveal that similar regions are explored by all the simulated systems (Figure 4b). Similar results were obtained by (i) the RMSD matrix of the binding groove of a reference simulation with respect to another (Appendix A) and (ii) by means of Zernike polynomials-based approach (see method). The latter, which essentially describes the pockets shapes, confirms the geometrical shape similarity for all the binding grooves (Appendix A). Contrary to our previous results [10], this might suggest a quite-rigid conformation of the pMHC pocket, which seems to be unaffected by the sequence of the peptide.

### 3.2. Peptide Interactions

The MD simulations also allow us to quantitatively investigate the interaction between the peptide and the two macromolecular units. The analysis of the hydrogen bonds (Table 2), involving the four peptides, confirms the presence of two anchor sites, located at the two ends of the peptides, which fix the ligand to the HLAs (Figure 6). In fact, the hydrogen bonds between the same residues of the binding groove and the residue P1-P2 and P8-P9 of the peptide are always maintained. Curiously, these h-bonds were found in our previous work on two HLA-B*27 subtypes [10], suggesting that, independently by the peptide sequence and the HLA type, the two ends of the ligand establish a similar interaction with conserved pocket residues. On the other hand, the hydrogen bonds between the peptide and the TCRs are specific of the complexes (Table 2 and Appendix A): in the case of the 1G4-ESO9C, the peptide forms one and three h-bonds with the alpha and beta chains. The mutations 4D-9V, in the ESO peptide, increase the number of h-bonds between the alpha and beta chains respectively. Moreover, the introduction of mutations in the CDR3α region in ILA1 determines the formation of a larger number of h-bonds, whereas the mutations in the CDR2β remove such bonds. However, in all the pMHC-TCR complexes the h-bonds formed between the peptide and the TCR are located in the central region (P4-P5) of the peptide. On the contrary, in the 1G4 simulations the solvent exposure per residue shows similar behavior for all the peptides (Appendix A).

### 3.3. Network and Cross-Correlation Matrix Analysis 

The cross-correlation (CC) maps show a main common coupling behavior between the main regions of the complexes (Figure 7). In fact, to a different extent, the Cα regions show an anticorrelation with the Cβ ones and a significant correlation between the α chain of the HLA and the Cα, Vβ, and Cβ is observed in all the systems. In the case of ILA1, the sequence changes do not affect the main correlation pattern, being that the extent of such correlation only slightly increased upon the mutations. Concerning the 1G4 systems, the modification of two a.a. residues in the peptide essentially decreases the extent of the motion correlation and anticorrelation.

As expected, the network analysis confirms that the regions of the TCR and pMHC directly interacting mainly contribute to the protein interaction network (Appendix A).

### 3.4. Interaction Energies

The interaction energies between selected regions of the complexes—involving the peptide and/or the binding groove—have been analyzed (Figure 8). As shown in Figure 8a, the distributions of such energy profiles all show favorable interactions between these regions and comparable values. The sole relevant difference was observed in the TCR-binding groove interaction, where the A*0201:hTERT-6A:ILA1 shows a sharp peak of the distribution in an energy range significantly higher than the other complexes. Interestingly, experimental data reports that no binding is observed in such a system. It is also worth noting that the single a.a. mutation provides a remarkable shift of the interaction energies as observed in both the 1G4 and ILA1α1β1 systems. The pocket electrostatic similarity computed by means of the Zernike Descriptor (Appendix A) shows a different electrostatic behavior of the pHLA-A*0201 interacting with ILA1. This might suggest that the TCR induces a modification of the electrostatic properties of the pMHC.

### 3.5. Membrane Behavior

To characterize the membrane properties, we measured the thickness of the lipid bilayers and the evolution of the distance between the two bilayers—hosting the pMHC and the TCR—along the trajectories. For the calculation of the membrane thickness (see methods for details), we compare the thickness variation between the two halves of each trajectory. As shown in Appendix A, no substantial differences were observed for such a property, i.e., the values are all within an interval of 3/4 nm. The distances between the two bilayers—computed by selecting the headgroup of each lipid (Appendix A) show a limited fluctuation around an average value.

## 4. Discussion

To characterize the conformational behavior of the interaction between the pMHC class I with the TCR, we performed an extended set of Molecular Dynamics simulations of five pMHC-TCR complexes, modelled in a heterogenous lipid environment. The chosen systems consist of the same MHC class I (HLA-A*02:01) binding to different peptides, which are recognized by three different TCRs: 1G4, ILA1 and ILA1α1β1. Such complexes differ from each other in the binding affinity reported in previous experimental studies, and all the systems present at least a mutation in the peptide or in the TCR sequence. In addition, we performed MDs of the unbound forms of the HLA-A*02:01, 1G4 and ILA1α1β1, to highlight the effects of the TCR interaction on the complex structural-dynamical behavior. To the best of our knowledge, this is the first computational study considering different pMHC-TCR complexes simulated in a lipid bilayer.

### 4.1. pMHC Influence on the TCRs Conformations

As expected, the presence of the pMHC affects the conformational behavior of the TCR in all the simulated systems. A more detailed analysis points out that this is verified in all the TCR regions, except for the Cβ ones which are quite unaffected by the presence of the HLA. The comparison between the different systems shows that the different TCR types can be discriminated by the explored conformations basin. In fact, the principal component analysis clearly demonstrates that the 1G4 and the ILA1α1β1 MD structures sample distinct regions of the conformational space, whereas the differences within the same TCR and different epitopes are quite limited. Very interestingly, the unique complexes showing no binding affinity for the A*0201:hTERT6A explores different regions of the conformational spaces at both global and local level.

### 4.2. TCR Effects on the Binding Groove Behaviour

Concerning the binding groove dynamics, our MD simulations point out that the MHC class I simulated in complex with different TCRs show that their conformational behavior is quite unaffected by both the TCR types and the sequence of the bound peptides. Such results were also confirmed by the RMSD and by the Zernike analysis, the latter providing a high pocket shape similarity in all the cases. On the other hand, some differences come out by the local analysis of the h-bonds between the TCR and the peptides. That is, the numerous h-bonds established between the ILA1α1β1and hTERT-6A are not found in the case of the ILA1 bound to the same peptides. Considering that the simulations have shown a limited conformational variability of the binding groove, we can argue that the TCR is the main determinant of the peptide interaction.

### 4.3. Dynamical Coupling of the pMHC-TCR

Although the kinetic assays reported differences in the binding activity, the pMHCs-TCRs exhibit similar interaction patterns. In detail, a remarkable correlation between the MHC alpha-chain and the whole TCR—which are not directly connected—was found in all the pMHC-TCR complexes. Moreover, the mutations of two residues in the peptide (ESO4D to ESO9C) induces a severe decrease of the correlation patterns, in both the TCR variable and TCR constant regions as well as in the MHC alpha-chain one. Such analysis suggests that a larger correlation between the variable and constant regions of the TCR does not necessarily imply an increased activation of the receptor. In fact, the 1G4-ESO4D, which experimentally shows a higher value of the K_D_ than the 1G4:ESO9C, presents a minor release of IFNγ [40] and a higher interchain correlation. In other words, a limited coupling between the pMHC and the TCR motions could imply a larger receptor response.

As expected, the interaction energies between the binding groove and the ILA1 T-cell receptor show higher values with respect to the other complexes, despite the very similar interaction energy distributions between the hTERT6A:ILA1 and the other peptide-TCRs. This might suggest that the complementarity between the MHC class I and the TCR is one of the key factors regulating this complex recognition process.

## 5. Conclusions

We performed several MD simulations to understand the contribution of each character involved in the TCR triggering. During the recognition process the MHCs adapt themselves to the TCRs surface, assuming similar conformations. On the other hand, different peptides induce specific changes in the TCRs structures: a single-residue mutation on the peptide determines conformational changes in all the TCR regions, excluding the *C*β one. Moreover, double-residues mutations are also responsible for a modification of the pMHC-TCR coupling motions. In addition, the TCR behavior seems to be linked to the its specific sequence: the TCR type determines changes in the variable regions and sequence alterations in the CDR2β and CDR3α variable regions (e.i ILA1α1b1) induce clear conformational changes especially in the *C*β region.

Accordingly, we argue that although the MHC is essential in the antigen presentation to the TCR, the activation process is mainly influenced by the peptide sequence, leaving the MHC conformational behavior quite unaffected. The ability of the TCR to modulate its response based on the peptide sequence/structure is then probably linked to the structural-dynamical behavior of the Vα, Vβ and Cα regions. Due to the time scale of the *C*β conformational rearrangements, (which should have a relevant role in the process [8]) we were able to detect modification only between radically different situations. i.e., binding vs. no binding. Despite the present limitations of the MD approach which are essentially due to a limited time scale with respect to the pMHC-TCR activation process, we think that our outcomes suggest a new key-reading of the role of each character involved in the pMHC recognition.

## Figures and Tables

**Figure 1 cells-09-00942-f001:**
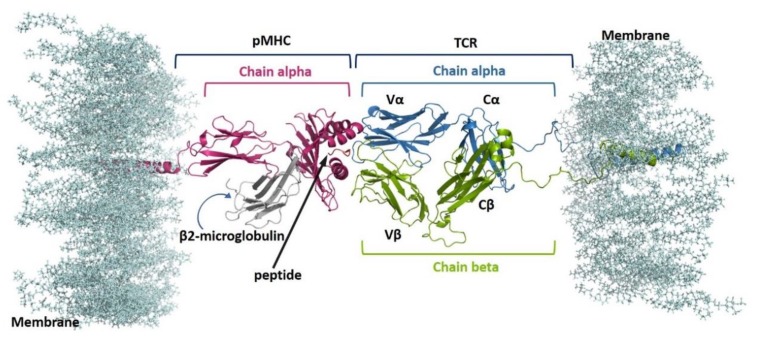
Model of one of the pMHC-TCR simulated complexes. The different regions are labelled: Vα/Vβ and Cα/Cβ refer to the TCR Variable regions (alpha/beta) and TCR Constant regions (alpha/beta), respectively.

**Figure 2 cells-09-00942-f002:**
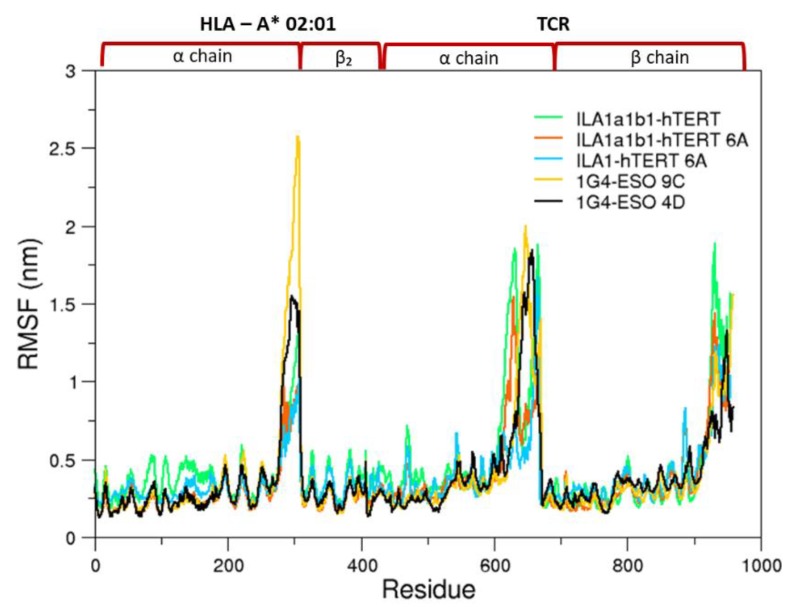
Root mean square fluctuation (RMSF) of the alpha carbon of the bound states. On top, the pMHC-TCR regions are reported.

**Figure 3 cells-09-00942-f003:**
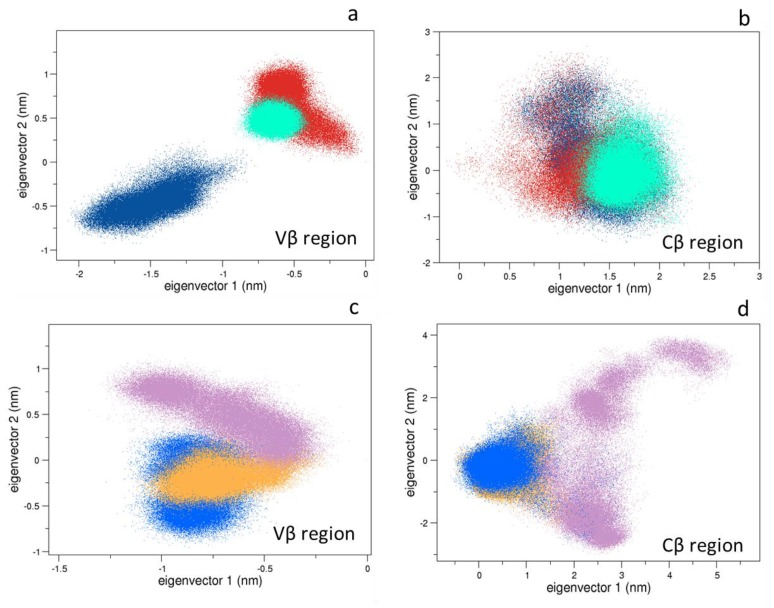
The 2D projections of bound-unbound states are shown. Panels (**a**,**b**): the projections of the MD trajectories of the 1G4 TCR on the 2D essential subspace (1G4-ESO9C in red, 1G4-ESO4D in cyan, unbound 1G4 in blue). Panels (**c**,**d**): the projections of the MD trajectories of the ILA1α1β1 TCR on the 2D essential subspace (ILA1α1β1-hTERT in orange, ILA1α1β1-hTERT-6A in blue, unbound ILA1α1β1 in mauve).

**Figure 4 cells-09-00942-f004:**
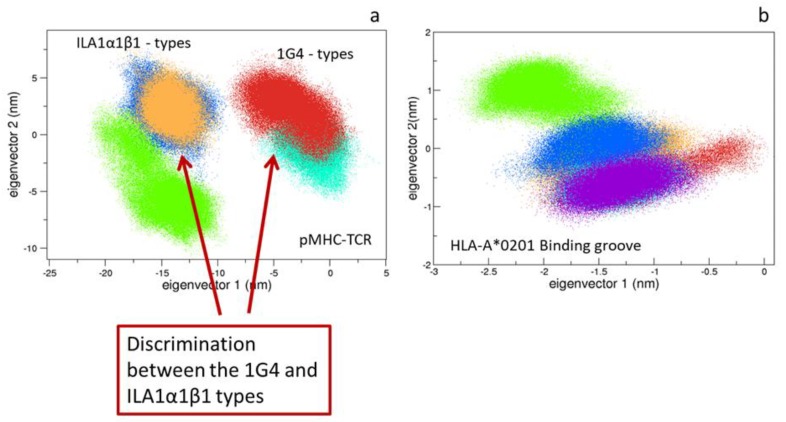
Comparison between the bound states. The projections on the common essential subspaces, defined by the two principal eigenvectors are shown. On the left (**a**), the 2D projections computed on the alpha carbons. On the right (**b**), the 2D projections of the Binding groove (1G4-ESO9C in red, 1G4-ESO4D in cyan, LA1α1β1-hTERT in orange, LA1α1β1-hTERT-6A in blue, ILA1 in green).

**Figure 5 cells-09-00942-f005:**
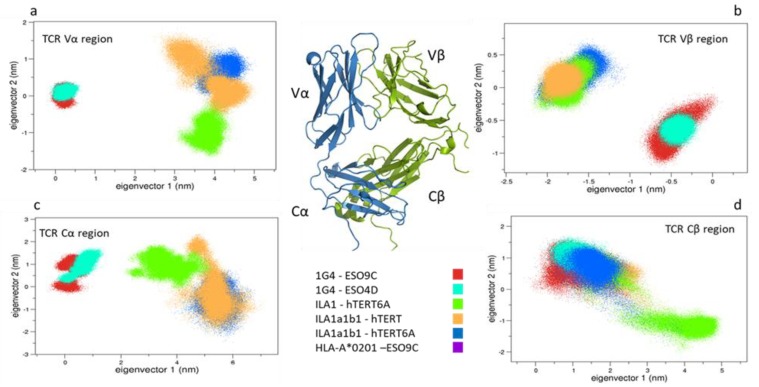
Comparison between the bound states. The projections of the TCRs Variable (**a**,**b**) and TCRs Constant regions (**c**,**d**) are reported.

**Figure 6 cells-09-00942-f006:**
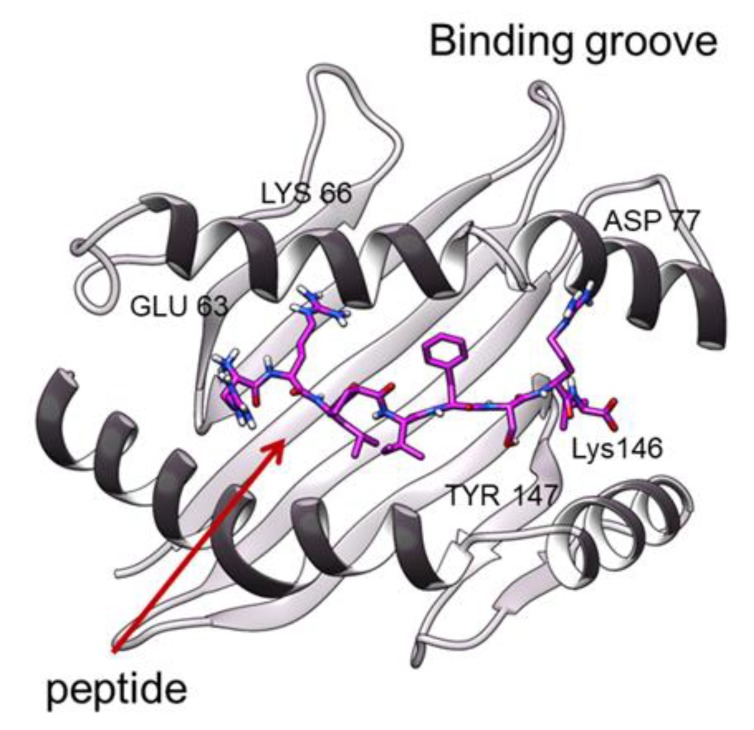
The Binding groove of a representative snapshot of 1G4-ESO9C. The residues making h-bonds are labelled.

**Figure 7 cells-09-00942-f007:**
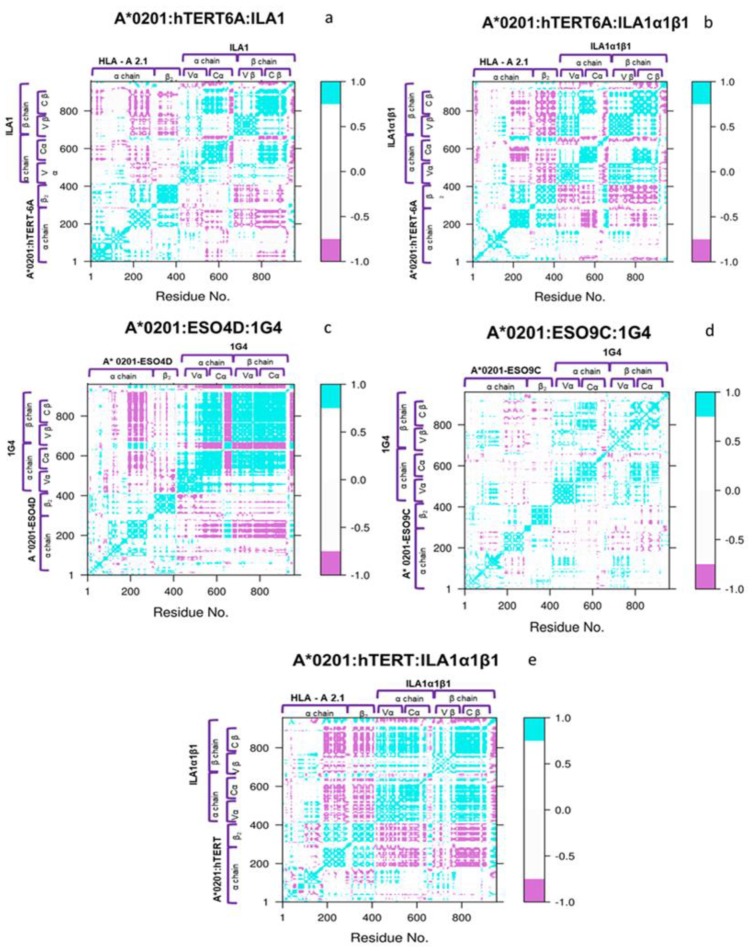
Cross correlation matrices. The correlation matrix of the whole complex pMHC-TCR is shown for the systems in study. The cyan regions indicate the presence of a correlation; the violet regions indicate an anti-correlation; the white regions show a non-correlation (**a**–**e**).

**Figure 8 cells-09-00942-f008:**
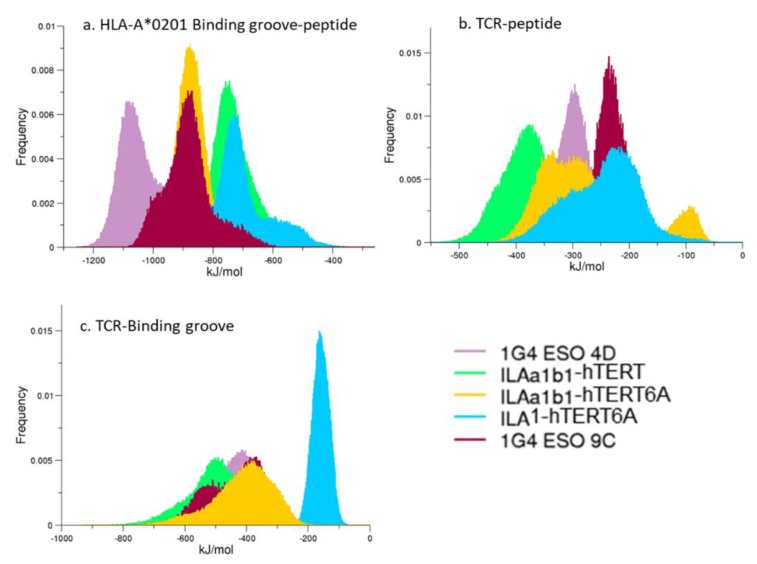
Interaction Energies. The distribution of the energy profiles of the complexes is reported. Panel (**a**): the interaction energy between the binding groove and the peptide; panel (**b**): the interaction energy between the TCR and the peptide; panel (**c**): the interaction energy between the TCR and the binding groove.

**Table 1 cells-09-00942-t001:** The simulated complexes and the corresponding experimental binding affinity data [41,42]. The MHC class I is the HLA-A*02:01. The last two lines report the sequence differences between the ILA1 and the ILA1α1β1 (the mutated residues are reported in red).

Complex with HLA-A*0201	Peptide Sequence	KD (uM)	Koff (s^−1^)	t ½ (s)	Kon (M^−1^ s^−1^)
1G4-ESO9C	SLLMWITQC	14	0.82	0.84	57 × 10^3^
1G4-ESO4D	SLLDWITQV	252	2.59	0.27	10 × 10^3^
ILA1α1β1-hTERT	ILAKFLHWL	0.002	1.6 × 10^−4^	4260	8 × 10^4^
ILA1α1β1-hTERT-6A	ILAKFAHWL	2	1.4 × 10^−2^	49	6 × 10^3^
ILA1-hTERT-6A	ILAKFAHWL	No binding			
**TCR**	**Mutation**	**CDR2β_49-54_**	**CDR3α_104-119_**		
ILA1	Wild type	SVGAGI	CAVDSATSGTYKYIFG		
ILA1α1β1	CDR2&3	SIHPEY	CAVDSATALPYGYIFG		

**Table 2 cells-09-00942-t002:** Hydrogen bonds between the peptide and the binding groove. The a.a. residues involved in the interaction are listed on the left. The residue of the peptide which makes the h-bonds is labelled with “P”, and the associate number denotes its position in the peptide sequence.

GROOVE	ESO9C: HLA A*0201	ESO4D: HLA A*0201	hTERT: HLA A*0201	hTERT-6A: HLA A*0201:ILA1α1β1	hTERT-6A: HLA A*0201: ILA1	ESO9C: HLA A*0201-Unbound
**Tyr 7**	**P1**	**P1**				**P1**
Glu 63	P1	P1	P1	P1	P1	P1
**Tyr 59**			**P1**			
Lys 66	P2	P2	P2	P4	P2/P4	
His 70			P3		P3	
Thr 73						P8
Asp 77	P8	P8	P8	P8		P8
Tyr 83				P3		
Tyr 84		P9	P9	P9		
Tyr 99	P2	P2	P2	P2	P2	P2
**Thr 143**	**P9**	**P9**	**P9**	**P9**	**P9**	**P9**
Lys 146	P9	P9	P9	P9	P9	P9
Trp 147	P8	P8	P8	P8	P8	P8
Tyr 159	P1	P1	P1	P1		P1
Tyr 171	P1	P1				P1

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
