# Peer review of "Molecular Dynamics Simulations Reveal Canonical Conformations in Different pMHC/TCR Interactions"

_cells, 2020, doi:10.3390/cells9040942_

Round 1

Reviewer 1 Report

The manuscript presents interesting results of an exhaustive study of computational simulation analysis by molecular dynamics of the binding behavior of five pMHC63 TCR complexes, modeled in a heterogeneous lipid bilayer. The information is valuable in the understanding of molecular aspects and details of the intermolecular interaction with the HLA in a hydrophobic environment. Nevertheless, some parts of the paper are confusing for the reader and in other insufficient information are offered. Therefore, careful major revision is necessary to make it more coherent and to clarify the benefit of this study. I do not recommend acceptance under the present form.

Major points

The manuscript needs of an extensive revision of Ingles grammar and spelling, some of which are noted in the text. The discussion is confusing, I suggest that the authors restructure the discussion by selecting themes by paragraphs.

Author Response

Response to Reviewer 1 Comments

The manuscript presents interesting results of an exhaustive study of computational simulation analysis by molecular dynamics of the binding behavior of five pMHC63 TCR complexes, modeled in a heterogeneous lipid bilayer. The information is valuable in the understanding of molecular aspects and details of the intermolecular interaction with the HLA in a hydrophobic environment. Nevertheless, some parts of the paper are confusing for the reader and in other insufficient information are offered. Therefore, careful major revision is necessary to make it more coherent and to clarify the benefit of this study. I do not recommend acceptance under the present form.

Point 1: Major points
The manuscript needs of an extensive revision of Ingles grammar and spelling, some of which are noted in the text. The discussion is confusing, I suggest that the authors restructure the discussion by selecting themes by paragraphs.
Response 1: We thank the referee for his/her comments. Accordingly, the grammar and spelling have been corrected. The discussion section has been revised and the conclusion section has been added to highlight
the major findings of the work.

Reviewer 2 Report

In the manuscript the authors investigate the conformational dynamics resulting from the interaction of the pMHC class I with the TCR through extensive molecular dynamics simulations of five complexes modelled in model membrane. Interestingly, results reported in the manuscript suggest a correlation between the conformational dynamics of the beta-chain constant regions of the TCS and experimentally measured T-cell activity, thus establishing a link between biological activity and protein dynamics. The simulations and methods are state of the art and the article is well-written. While I would consider this manuscript more appropriate for another journal of the MPDI group (e.g. Biomolecules or Int. J. Mol. Sci.) I think it can be of interest to a more biology-oriented audience like that of Cells. I would recommend publication after the following points have been addressed:

  1. The data analysis depends critically on the application of principal component analysis. As it stands the information on the method provided in the manuscript is clearly not satisfactory. To broaden the readership and scholarship of the manuscript the authors should provide a brief primer of PCA and indicate some studies where it has been used to explore conformational dynamics. A relatively recent example on protein folding that should be cited is https://doi.org/10.3390/ijms140917256
  2. The equations 1 and 2, seem to have been copied from latex or something similar. They should be correctly edited and displayed.
  3. The conclusions should not mention figures and SI.
  4. Caption of Figure 3. I can’t find any cyan in the plots on the top row.

Author Response

Response to Reviewer 2 Comments

In the manuscript the authors investigate the conformational dynamics resulting from the interaction of the pMHC class I with the TCR through extensive molecular dynamics simulations of five complexes modelled in model membrane. Interestingly, results reported in the manuscript suggest a correlation between the conformational dynamics of the beta-chain constant regions of the TCS and experimentally measured T-cell activity, thus establishing a link between biological activity and protein dynamics. The simulations and methods are state of the art and the article is well-written. While I would consider this manuscript more appropriate for another journal of the MPDI group (e.g. Biomolecules or Int. J. Mol. Sci.) I think it can be of interest to a more biology-oriented audience like that of Cells. I would recommend publication after the following points have been addressed:

Point 1: The data analysis depends critically on the application of principal component analysis. As it stands the information on the method provided in the manuscript is clearly not satisfactory. To broaden the readership and scholarship of the manuscript the authors should provide a brief primer of PCA and indicate some studies where it has been used to explore conformational dynamics. A relatively recent example on protein folding that should be cited is
https://doi.org/10.3390/ijms140917256
Response 1: We thank the referee for his/her comments. A more detailed explanation of the use of Principal Components Analysis have been added.

Point 2: The equations 1 and 2, seem to have been copied from latex or something similar. They should be correctly edited and displayed.
Response 2: The equations 1 and 2 have been corrected.

Point 3: The conclusions should not mention figures and SI.
Response 3: We removed the mention to the figures and to the SI in the conclusion section.

Point 4: Caption of Figure 3. I can’t find any cyan in the plots on the top row.
Response 4: The figures have been checked

Reviewer 3 Report

In this manuscript, the authors use atomistic molecular dynamics simulations to investigate a key step in the activation of cytotoxic T lymphocytes in the context of the immune response. In particular, they study detailed conformational changes in complexes formed by different T-cell receptors and the same major histocompatibility complex. The authors succeed in establishing correlations between a quantitative analysis of the conformations explored in their different simulations and known experimental data. The manuscript is technically sound and the experimental design solid. The simulations are state-of-the-art and sampling is adequate. The data justify the conclusions. I have two main issues with the paper. First, it would be nice to distil a mechanistic picture from the rich data described in this manuscript. Starting from the beginning, the authors should make an effort to nail down a more precise set of questions already in the introduction. Then in the discussion, I would like to see a concise mechanistic picture that puts the conclusions of the paper in the context of the phenomenon that the authors are investigating. What do we learn from all these simulations? How is this relevant to understand the activation of T-cells during immune response? Adding these considerations would make the paper not only more interesting for a wider audience but also more fun to read. Second, although the authors comment at the end of the manuscript about the importance of the membrane and a detailed modelling of it, funnily the role of the lipid bilayer never appears in any analysis or discussion. What are the lipids doing? Is there anything interesting happening to the transmembrane domains in the two bilayers? Is the distance between the two bilayers evolving? Does it correlate interestingly with the observed conformational changes? In this context, I find the lipid compositions chosen by the authors (POPC 90% PIP2 7% POPS 3%) rather odd! If I understand correctly, this should be a model of the plasma membrane. If this is true, then I do not understand why cholesterol is completely missing, and why PIP2 is so overrepresented (I am using J. Am. Chem. Soc. 2014, 136, 41, 14554-14559 as a reference). It would be opportune if the authors commented about their choice of lipids. Minor observations - In the abstract, the density of acronyms is really high and a bit uncomfortable. For instance, is (p; line 21) for epitope recognition really necessary? - Line 24 Triggering of what? - In the introduction some acronyms are introduced again and some are not. I would ask to be consistence, by introduction all acronyms anew. - Line 63. Modelized -> Modelled - Figure 1 legend. The figure can be made clearer. Please indicate which side is the TCL one and which is the other. Please explain in the legend what HLA-A*02:01 is. - Eq. 1 and 2 could be properly displayed!

Author Response

Response to Reviewer 3 Comments

In this manuscript, the authors use atomistic molecular dynamics simulations to investigate a key step in the activation of cytotoxic T lymphocytes in the context of the immune response. In particular, they study detailed conformational changes in complexes formed by different T-cell receptors and the same major histocompatibility complex. The authors succeed in establishing correlations between a quantitative analysis of the conformations explored in their different simulations and known experimental data. The manuscript is technically sound and the experimental design solid. The simulations are state-of-the-art and sampling is adequate. The data justify the conclusions. I have two main issues with the paper.

Point 1: First, it would be nice to distil a mechanistic picture from the rich data described in this manuscript. Starting from the beginning, the authors should make an effort to nail down a more precise set of questions already in the introduction. Then in the discussion, I would like to see a concise mechanistic picture that puts the conclusions of the paper in the context of the phenomenon that the authors are investigating. What do we learn from all these simulations? How is this relevant to understand the activation of T-cells during immune response? Adding these considerations would make the paper not only more interesting for a wider audience but also more fun to read.
Response 1: We thank the referee for his/her comments and suggestions. Accordingly, we have now revised the abstract and the introduction section to clarify the questions which we were able to address in the manuscript. Moreover, a modified discussion section and a conclusion section have been added to clarify the major findings, trying to link our results to the TCR response behaviour.

Point 2: Second, although the authors comment at the end of the manuscript about the importance of the membrane and a detailed modelling of it, funnily the role of the lipid bilayer never appears in any analysis or discussion. What are the lipids doing? Is there anything interesting happening to the transmembrane domains in the two bilayers? Is the distance between the two bilayers evolving? Does it correlate interestingly with the observed conformational changes?
Response 2: We agree with the referee. However, we would like to point out that the importance of including the membrane is that a realistic sampling of the conformational behavior of our system is achievable in the presence of the membrane only. That is, the presence of the membrane influences the structural-dynamics behavior of pMHC-TCR complex. On the other hand, the referee is right about the analysis on lipids, which were missing in the previous version of the manuscript. Accordingly, we have now added the suggested analysis to the manuscript.

Point 3: In this context, I find the lipid compositions chosen by the authors (POPC 90% PIP2 7% POPS 3%) rather odd! If I understand correctly, this should be a model of the plasma membrane. If this is true, then I do not understand why cholesterol is completely missing, and why PIP2 is so overrepresented (I am using J. Am. Chem. Soc. 2014, 136, 41, 14554-14559 as a reference). It would be opportune if the authors commented about their choice of lipids.
Response 3: We thank the referee for his/her comment. Although cholesterol is important to maintain integrity and mechanical stability in the eukaryotic plasma membrane, recent findings reported that it can alter the T cell activation. Because our goal was to detect conformational differences related to the activation process, we did not include the cholesterol in our model, to avoid any kind of artefacts due to the TCR-cholesterol interaction [ Wang et al., 2016; Swamy et al., 2016]. Moreover, the lipid composition chosen (POPC 90% PIP2 7% POPS 3%) is based on the work of Chavent et al., and Zech et al. to approximate the lipid composition of a mammalian cell membrane
[van Meer et al., 2008]. These studies underline the role of PIP2 in the TCR activation process. Accordingly, the following sentences have been added to the “Materials and method” section: “Such a lipid composition is based on the work of Chavent et al. [14] and Zech et al. [15] to approximate the lipid composition of a mammalian cell membrane [16]. These studies reported an essential role of the PIP2 in receptor activation and TCR triggering. Note that recent findings reported that cholesterol can inhibit T cell activation and thus, this lipid was not included in our membranemodel [17-18].”

Point 4: Minor observations - In the abstract, the density of acronyms is really high and a bit uncomfortable. For instance, is (p; line 21) for epitope recognition really necessary? - Line 24 Triggering of what? - In the introduction some acronyms are introduced again and some are not. I would ask to be consistence, by introduction all acronyms anew. - Line 63. Modelized -> Modelled -

Figure 1 legend. The figure can be made clearer. Please indicate which side is the TCL one and which is the other. Please explain in the legend what HLA-A*02:01 is. - Eq. 1 and 2 could be properly displayed!
Response 4: The referee minor points have been addressed.

Round 2

Reviewer 1 Report

My suggestions were fully answered. The manuscript is ready to be published.